# Characterization of Sugar Reduction in Model Confectionary Gels Using Descriptive Analysis

**DOI:** 10.3390/gels8100644

**Published:** 2022-10-11

**Authors:** Elle McKenzie, Youngsoo Lee, Soo-Yeun Lee

**Affiliations:** Department of Food Science and Human Nutrition, University of Illinois Urbana-Champaign, 905 South Goodwin Ave., 486A Bevier Hall, Urbana, IL 61801, USA

**Keywords:** descriptive analysis, sugar reduction, gelatin, carrageenan, konjac glucomannan

## Abstract

Successful sugar reduction in food products mimics the sensory and functional properties of the full sugar counterpart. The initial step of sugar reduction is to determine how the absence of sugar affects these properties. Descriptive analysis was conducted on four gel types (gelatin, ι-carrageenan, κ-carrageenan, and konjac glucomannan) and a range of sugar concentrations from 0–20% *w*/*v* to create a sensory profile of model confectionary gels for comparison to instrumental texture profile analysis data. The sensory descriptive data were analyzed using analysis of variance and principal component analysis. Correlation analysis, cluster analysis, and partial least squares regression (PLS-R) were used to compare and correlate sensory and instrumental data. Regardless of sugar concentration, sensory analysis primarily clustered samples by gelling agent type, such as in the case of konjac glucomannan consistently being characterized as chewy. Cohesion and gumminess were correlated highly with melt-in-mouth and a jiggly texture, while adhesion and fracturability were negatively correlated. In the PLS-R samples biplot, gelatin and iota carrageenan samples were located near these attributes indicating their aptness as descriptors. In conclusion, descriptive analysis provided a more discriminating method for characterizing model confectionary gels.

## 1. Introduction

Overconsumption of added sugar and high sugar products can lead to numerous health problems that result from the consequential weight gain, such as cardiovascular disease [1] and type 2 diabetes mellitus [2]. As a result of these negative effects on consumer health, many customers have looked for healthier alternatives to their high sugar products. Food companies have begun responding to these market trends by investigating sugar reduction methods and producing reduced sugar or sugar-free alternatives to their products.

The main ways to accomplish sugar reduction can be as simple as removing sugar from a product’s formulation or as complex as a complete formulation change in which added functional ingredients are utilized [3,4,5,6]. However, by modifying a product in such a way comes the challenge of maintaining both its functional and sensory aspects. Sugar plays an essential functional role in many products, providing bulk, modifying viscosity, and contributing to preservation [7,8,9]. Sensory perception of reduced sugar products is also affected. For example, sugar often provides a different taste profile than sugar alternatives, which often have licorice or metallic aftertastes. While there are many roles that sugar holds in product development, one of the main effects of sugar reduction are modifications in mouthfeel and texture, which alter both the instrumental and sensory results [10,11]. To quantify these textural variations, instrumental methods have traditionally been implemented for fast and cheap analysis; however, there is concern over instrumental data correlation to the sensory perception, and consequently, the consumer perception.

For gel systems, the Bloom test, an instrumental method, is often conducted to determine a single measurement of Bloom strength or more simply described as a form of hardness. While Bloom strength is a valuable measurement in analyzing product’s texture, it does not represent the overall mechanical behavior or texture, such as elasticity or other time-based attributes, which plays a crucial role in their textural perception. In order to obtain a more comprehensive textural profile, texture profile analysis was conducted to analyze parameters of the gel systems. 

Texture profile analysis (TPA) is one of the main instrumental methods for comparison to sensory attributes generated through descriptive analysis and was originally created by a group of scientists at the General Food corporation [12]. As an imitative rheology method, TPA characterizes the texture of food products through a two-bite compression cycle under control conditions. The recognized TPA parameters and their sensory definitions are defined in Appendix A.

Other works have utilized this method to characterize gel systems, such as Yusof et al., 2019 [13], Chandra et al., 2015 [14] and Rahman M et al., 2009 [15]. In these studies, TPA was the main technique to characterize changes in texture based on different processing methods and gelling agent sources, specifically in gelatin. While there are studies which instrumentally characterize the changes in gel systems when sugar reduction occurs [16,17,18], they usually lack a sensory component, and because the correlation between instrumental and sensory data is dependent on the matrix, more research is required to characterize sugar reduction in individual food products, such as gels.

A variety of works can be found in the literature comparing instrumental and sensory method [19,20,21,22,23], although less work has been done regarding the aspect of sugar reduction [24,25], especially in the area of confectionary gels. Developing a correlation between sensory and instrumental results is critical for product development and the success of sugar reduction in products. The consumer or sensory perception of a product is often the final determinant of hedonic results. This research works to characterize and highlight the changes in sensory perception that occur during sugar reduction in gel-like products and correlate that to instrumental data. This sensory and correlation data can be used as reference for future work when initially attempting to reduce sugar in gel-like products. The objectives of this research were to investigate the effects of sugar reduction on model confectionary gels made from various gelling agents, some which are under characterized in the literature, and to correlate and compare sensory and instrumental measurements to determine which attributes best describe the model confectionary gels.

## 2. Results and Discussion

### 2.1. Sensory Analysis

Attributes, definitions, and references generated for the sensory profile of the model confectionary gels during the descriptive analysis are detailed in Table 1. Results from the ANOVA of 13 sensory attributes are summarized in Table 2. Panelists were a significant source of variance (*p* < 0.001) for all attributes; however, this is typical for descriptive analysis outcomes, indicating that panelists were using the scales differently. There were not significant replication effects seen for any of the attributes demonstrating the reproducibility of the panel. There was a presence of panelist by replication effect for the taste attributes sweet and bitter, indicating that there were disagreements between panelists about which replication had higher intensity of a given attribute. This could be due to differences in perception of taste among panelists. There were a number of significant panelist-gel type and panelist-sugar concentration interactions, indicating poor alignment among panelists for some attributes. After an adjusted F-test using a mixed-model ANOVA was conducted treating the panelists as a random effect, attributes that were significantly different by either gel type or sugar concentration were included in the PCA.

Of the 13 generated attributes, the following terms were significantly different based on gel type: cloudy, glossy, bubble, fishy aroma, metallic aroma, sweet, sweet aftertaste, jiggly, creamy, chewy, melt-in-mouth, and mouthcoating. In comparison, only six attributes were significantly different based on sugar concentration: cloudy, sweet, bitter, sweet aftertaste, jiggly, and chewy, which are mainly focused on the taste and aftertaste of the samples. The attributes generated for the model gel confections are similar to the findings in the literature for semi-solid foods and gel desserts, specifically the attributes sweet, creamy, and jiggly [26,27,28,29,30]. Additionally, many of the attributes can be categorized under primary parameters that are correlated to texture profile analysis [31].

Principal component analysis was conducted on the significant attributes for both gel type and sugar concentration (Figure 1 and Figure 2). The biplot of PC1 vs. PC2 from the PCA of the matrix of significant sensory attributes across model confectionary gel samples based on gel type is shown in Figure 1. The PCA explained 74.94% of the variation in which PC1 and PC2 explained 52.58% and 22.36% of variance, respectively. The model confectionary gels were clustered together on the PCA biplot by gel type, regardless of sugar concentration. Gelatin samples were characterized by glossy, sweet, and sweet aftertaste. These descriptors align well with the known properties of gelatin for being colorless and tasteless [32]. In these samples, sweetness from the sugar was the main taste perceived as opposed to bitter that was perceived in other gelling agents. ι-carrageenan samples were characterized by mouthcoating and creamy. In the food products, such as dairy or plant-based alternatives, where a creamier texture is desired, carrageenans are often used as the texturizer. This finding is also supported by the evidence of texture of IC being softer than its KC counterpart [26,33,34,35]. Konjac glucomannan samples were highly correlated with cloudy, bubbly, and chewy, while κ-carrageenan samples were not explained well by this biplot. There is a clear distinction among gel types, as certain attributes were highly correlated with certain gel types, which helped to define the unique sensory profile of each gelling agent.

The biplot of PC1 vs. PC2 from the PCA of the matrix of significant sensory attributes across model confectionary gel samples based on sugar concentration is shown in Figure 2. The PCA explained 88.50% of the variation in which PC1 and PC2 explained 50.76% and 37.74% of variance, respectively. Regardless of the sugar concentration, konjac glucomannan samples were highly characterized by chewy and cloudy. Most gel samples without sugar were characterized by bitter taste. Without the masking effect of sugar, the bitterness of the samples was more easily perceived by panelists. Surprisingly, the samples with the highest sugar concentration did not correlate with a sweet or sweet aftertaste, indicating that with the addition of sugar, other factors, such as the textural attributes, better explained the samples. Texture is a main determinant in the perception of sweetness and flavor release in products, as samples with the same sugar concentration can be perceived with different intensities of sweetness depending on hardness and fracturability [36,37]. Similar results were found in studies which attempted to replace sugar with alternative sweeteners in jam-like products. Akesowan and Choonhahirun (2019) [36] found that “harder” gels were perceived as less sweet than their softer counterparts, due to the limited flavor release.

Cluster analysis was conducted on the significant attributes for gel type and sugar concentration. In both, the dendrogram for gel type and the dendrogram for sugar concentration, three major clusters were identified, and generally the gel types are clustered together, regardless of sugar concentration (Figure 3 and Figure 4). This is similar to the findings of Suebsaen et al. (2019) [38], where the gelling agent used to formulate banana gel desserts had a significant impact on the overall texture and sensory attributes of the samples, regardless of other ingredients. Based on gel type (Figure 3), gelatin and KC are clustered together, while the samples of KGM and IC are clustered into their own separate branches. This indicates that based on sensory attributes significant to gel type, gelatin and KC were the most similar in terms of sensory perception. This may be due the gelling mechanism of KC, which forms much larger strands of helices compared to other types of carrageenan, similar to the gelling mechanism of gelatin, which also relies on helical aggregation [32,33,39,40,41]. 

Within the smaller branches of the dendrogram based on gel type (Figure 3), the higher levels of sugar concentration (10–20% *w*/*v*) were clustered together first within a gel type, indicating that the panelists were able to discriminate across the levels of sugar concentration within gel types [42]. Based on sugar concentration (Figure 4), all KGM samples were clustered on their own, while the other gel types are clustered together in a larger group. The only cluster that is not characterized by a specific gel type but rather by sugar concentration are the 0% (*w*/*v*) sugar samples, demonstrating that there was a perceived sensory difference in sugar concentrations based on the attributes of cloudy, sweet, bitter, sweet aftertaste, and chewy. In conjunction with the PCA data and the masking effect of sucrose Mennella, 2015, it is likely that bitterness is the main differentiating characteristic. 

### 2.2. Relationship between Sensory Attributes and Instrumental Parameters

Cluster analysis of the instrumental parameters of the 20 model confectionary gels provided a comparison to how samples were clustered using sensory attributes (Figure 5). With sensory data based on gel type, gelatin samples were clustered with KC, but the further branches were distinguished by individual gel type. In comparison, sensory data based on sugar concentrations clusters gelatin with both types of carrageenan, and samples with no sugar are separated in their own cluster. However, when the gel samples are clustered using instrumental parameters, gelatin is grouped with only IC with there being little distinction between gel type. This perceived difference between sensory and instrumental measurements demonstrates the ability of sensory data to be more discriminative in characterizing samples, which has been exhibited in the literature [11]. In the work conducted by Kappes et al. (2006) [11], panelists were able to detect mouthfeel differences between diet and regular carbonated beverages, despite the instrumental results being minute. The sensory data were able to cluster samples consistently by gel type and to some extent sugar concentration. The more sensitive differentiation through sensory methods may also be aided by the use of other sensory modalities, such as appearance, smell, and taste, when creating the sensory profile, while the instrumental measurements used in this study only characterized texture of the samples.

The correlation matrix of the mean sensory texture attributes and instrumental parameters is shown in Table 3. Hardness is most negatively correlated with creamy (−0.72) and then mouthcoating (−0.66). Similarly, fracturability was negatively correlated with most sensory texture attributes, including creamy (−0.56), mouthcoating (−0.60), and melt-in-mouth (−0.54); however, it was positively correlated with chewy (0.71). Many of the textural sensory attributes that are negatively correlated to hardness and fracturability define a product that has resistance to bite [43,44]. This may characterize creamy, mouthcoating samples as softer and less elastic, and could be a key factor in adjusting the texture of a product. Cohesion was correlated best with sensory attributes melt-in-mouth (0.77) and then jiggly (0.71), but negatively correlated with chewy (−0.76). Melt-in-mouth and jiggly were also positively correlated with gumminess, the product of cohesion and hardness parameters (r = 0.65 and 0.73, *p* < 0.05). Adhesion was best correlated with chewy (0.58), but negatively correlated with both jiggly (−0.73) and melt-in-mouth (−0.68). These correlations between instrumental and sensory attributes allow for more efficient characterization through instrumental methods. However, more insight is needed to translate the instrumental parameters to human perceptions of food, especially for fracturability, which had a high correlation with a number of the sensory attributes. This aligns with other studies, such as Garcia Loredo and Guerrero (2011) [22], which looked at the correlation between instrumental and sensory ratings. The researchers found that hardness and fracturability had higher correlation with their TPA equivalents. 

In addition to correlation between instrumental and sensory texture attributes, some attributes within the methods were highly correlated to each other. For example, creamy and mouthcoating, both sensory attributes, were highly correlated with each other (0.94). Melt-in-mouth and chewy were negatively correlated (−0.94), while melt-in-mouth was positively correlated with jiggly (0.94). Fracturability and hardness were strongly correlated (0.88). This is due to the similarity in their calculation from TPA data based on rupture point of either the surface or internal structure. While adhesion was strongly negatively correlated with both gumminess (−0.96) and cohesion (−0.87), which is caused by adhesion and cohesion being opposing internal forces. Many of the correlations were similar to the findings of Daget and Collyer (1984) [45] which compared quantitative descriptive analysis to physical measurements for different gel systems, including gelatin, pectin and κ-carrageenan, but not across sucrose concentrations [45].

Partial least squares regression (PLS-R) was conducted on the instrumental and sensory texture attributes to relate the measurements, and a biplot was created (Figure 6). The biplot explained 56.9% of the data. It can be seen that cohesion was mostly explained by jiggly and melt-in-mouth sensory attributes, which was consistent with the findings from the correlation matrix. The other TPA measurements did not appear to explain any of the other sensory attributes. In the samples biplot, most samples were grouped by gel type with konjac glucomannan samples being characterized by hardness, fracturability, and chewy. The ι-carrageenan samples with 5–15% (*w*/*v*) sugar are characterized by creamy and mouthcoating, similar to what was found in the PCA biplot. Some gelatin and ι-carrageenan (0 and 20% *w*/*v*) samples were clustered near to where gumminess and jiggly are located. κ-carrageenan samples were located near adhesion; however, it was not characterized by any other sensory attributes. 

Based on both instrumental and sensory results, both carrageenan samples in some ways were perceived similarly to gelatin samples. Interestingly, the ι-carrageenan samples that were most similar to gelatin were the ones with 0 and 20% (*w*/*v*) sugar concentration based on PCA data (Figure 2). However, based on cluster analysis of sensory attributes which considered more modalities than texture (Figure 3), gelatin is clustered with KC, so it is unclear which carrageenan has the most similar texture or sensory profile to gelatin. 

## 3. Conclusions

Among the gel samples, most were characterized and clustered by their gel type regardless of sugar concentration. This emphasizes when reducing sugar in a product, the selection of gelling agent has a more significant impact on sensory perception than the sugar concentration. These findings indicated that slight sugar reduction is possible in gel models without a significant impact on the texture perception. The choice of gelling agent is more important in achieving a desired texture. Gelatin samples were perceived as gummy and glossy, and IC samples were perceived as creamy and mouthcoating. In contrast, KC samples were not well described by either sensory or instrumental measurements. KGM samples were overall clustered differently from other gel types, and perceived as chewy, hard, and cloudy. Cluster analysis of the descriptive analysis data was able to differentiate the gel type and discriminate a difference in samples that contained no sugar when clustered based on significant attributes for sugar concentration.

It can be concluded that sensory attributes provided a better assessment of differentiating between gel types and sugar concentrations as a trained panel was able to provide better discrimination across samples. This study highlights the importance of sensory perception in products, specifically when looking at different gelling agents. However, in the experimental design, no additional sweeteners were used to mitigate the difference in sweetness among samples with sugar reduction. Additionally, since model gels were used, more information would be needed to determine the effect of sugar reduction in more complex products with flavoring and colorants. Future work could focus on characterizing a wider variety of gel types as well as more complex formulations through the use of descriptive analysis. Additionally, a consumer test on the model confectionary gels could be conducted to determine the drivers of liking.

## 4. Materials and Methods

### 4.1. Raw Materials and Sample Preparation

Gelatin (Rousselot, Inc; Mukwonago, WI, USA), ι-carrageenan (TICALOID^®^ 100), κ-carrageenan (TICALOID^®^ 710H) (Ingredion, Inc; Westchester, IL, USA), and konjac gum (Modernist Pantry; Eliot, ME, USA) were used as gelling agents in formulating the gels for both instrumental and sensory testing. The gelatin powder had bloom and grain size of 250 PS 30. Calcium hydroxide (Modernist Pantry; Eliot, ME, USA) was used as a deacylating agent in the preparation konjac gum gels. Treatments containing sucrose utilized granulated sugar from Domino Foods, Inc (Yonkers, NY, USA). There was no further processing of materials before use in sample preparation.

Gelatin (G) gels (2.1% *w*/*v*) were created by adding the gelling agent to room temperature deionized water and then heated to 45 °C. Once heated, the solution was poured into 60 mL straight walled cylindrical molds with a diameter of 45 mm and a height of 45 mm. ι- and κ-carrageenan (IC and KC) gels (2.3 and 1.2% *w*/*v*) were made by adding the gelling agent to room temperature deionized water and allowing the powder to swell for one hour. Once the mixture was homogenous, it was heated to 85 °C. Once heated, the solution was poured into the cylindrical molds. Konjac glucomannan (KGM) gels (1.6% *w*/*v*) were created by adding the gelling agent powder to 90% room temperature deionized water. Calcium hydroxide was added to the mixture in a ratio of 2% (*w*/*w*) of KGM powder and then into the remaining 10% deionized water before being added to the final mixture. The mixture was then allowed to swell for one hour and placed in the cylindrical molds, vacuum sealed and heated at 85 °C in a water bath for two hours. All gels were allowed to cool to room temperature before storing in the fridge at 4 °C. Gelling agent concentrations were selected which maintain a consistent initial hardness from texture profile analysis data in samples with no sugar. If sugar was added to samples, it was added simultaneously with the gelling agent. Sugar concentrations were 0, 5, 10, 15, and 20% *w*/*v* for each gelling agent. Overall, 20 samples were tested.

### 4.2. Sensory Evaluation

#### 4.2.1. Participants

Nine subjects (8 female, 1 male, age range 18–24 years) participated in the study and were recruited from the University of Illinois at Urbana-Champaign student body based upon interest and availability. Panelists were untrained in sensory experiments before the study. They were advised not to eat or drink or smoke at least 30 min prior to a session. Panelists were screened with a basic taste test, in which they were asked to identify six basic tastes with a correct identification of at least three. Additionally, a 6-n-2-propylthioracil (PROP) taster test was conducted to screen out non-tasters; thus, all panelists were PROP tasters, as it is generally recognized that non-tasters are less sensitive to some perceptions compared to tasters.

#### 4.2.2. Basic Taste Solutions and PROP Preparation

Sucrose (0.07% *w*/*v*, Domino Foods, Inc., Yonkers, NY, USA), sodium chloride (0.1% *w*/*v*, The Great American Spice Co., Fort Wayne, IN, USA), citric acid (0.05% *w*/*v*, Hearthmark, LLC, Fishers, IN, USA), monosodium glutamate (0.015% *w*/*v*, Ajinomoto North America, Inc., Itasca, IL, USA) and caffeine (0.024% *w*/*v*, MP Biomedicals, LLC, Solon, OH, USA) solutions were prepared with filtered water at levels above their respective basic taste threshold. The solutions were presented in 29.5 mL plastic cups with 20 mL in each container. The solutions were given to the panelists in addition to a blank of filtered water. 6-n-2-propylthioracil (PROP) taster status was determined using PROP filter paper (Sensonics International, Haddon Heights, NJ, USA).

#### 4.2.3. Sample Preparation

Gel samples for sensory testing were prepared as cited in the sample preparation section and served in 29.5 mL plastic cups with lids (Dart Container Corporation, Mason, MI, USA). Samples were taken out of the refrigerator at least two hours prior to evaluation to equilibrate to room temperature. Plastic spoons were provided to handle the samples. The plastic serving cups were labeled with random 3-digit codes.

#### 4.2.4. Panelist Training 

During an introduction session, panelists were first given a presentation on the basics of sensory evaluation to assist in their understanding of the procedures. Six sessions were used for term generation and reference identification. The terms and references were generated and chosen through an iterative process based on their ability to describe gel samples. Reference intensities were iteratively calculated as the average intensity rating of the group over the course of training. During these sessions, a rinse protocol of room temperature water was also established. The training sessions consisted of ten group sessions, two individual practice sessions, and five individual evaluation days.

A randomized complete block design with two replications was conducted. Panelists rated samples in individual booths with incandescent lighting at 21 °C. Eight samples were presented during each evaluation session with a 5 min pause after four samples to reduce fatigue. The samples were presented monadically with random 3-digit codes. The panelists were instructed to rinse and expectorate between each sample with room temperature water provided to them in 473 mL cups. In the booth, panelists were given a sheet with the attributes and their definitions, and references and their intensities. The physical references were available outside the booth for panelists to familiarize themselves with before or during the session. An online ballot (Qualtrics, Provo, UT, USA) was created with unstructured lines scales (0 to 15) and reference intensities labeled in the question.

### 4.3. Texture Profile Analysis

For texture profile analysis, samples were tested in 60 mL cylindrical molds to stabilize the samples. A TA.XT2 Texture Analyzer instrument (Stable Food Micro Systems, Texture Technologies Corp; London, UK) was utilized with a 7 mm diameter probe. Each treatment was tested in triplicate with two instrumental repetitions. The preset speed was 1 mm/s, the test speed was 2 mm/s, and the post-test speed was 5 mm/s. A 75% compression was conducted with 5 seconds between “bites”. The trigger force was set to 3 g, and a 5 kg load cell was used. Hardness, cohesion, adhesion, fracturability, and gumminess were extracted from the data based on definitions from Bourne (1978) [12] as presented in Appendix A.

### 4.4. Statistical Analysis

Data from both instrumental and sensory methods were analyzed using XLStat (Addinsoft, New York, NY, USA). Analysis of Variance (ANOVA) was conducted with Fisher’s least significant difference (LSD) mean separation to determine if there were differences among gel type, sugar concentration, panelists, replications, or interactions for each attribute. If panelist by gel type or sugar concentrations interaction was significant for a given attribute, an adjusted F test (mixed-model ANOVA) was performed. Principal components analysis (PCA) by correlation matrix was used to identify terms that best described each sample for gel type and sugar concentration. Correlation analysis was used to determine the relationship between instrumental and sensory attributes. Cluster analysis was conducted to assess how each set of attributes (instrumental vs. sensory) classified samples, and dendrograms were created using the Pearson correlation coefficient for similarity. Partial least squares (PLS) regression was conducted to relate instrumental and sensory data matrices with instrumental parameters used as the explanatory/independent variables (X) and sensory attributes used as the response/dependent variables (Y).

## Figures and Tables

**Figure 1 gels-08-00644-f001:**
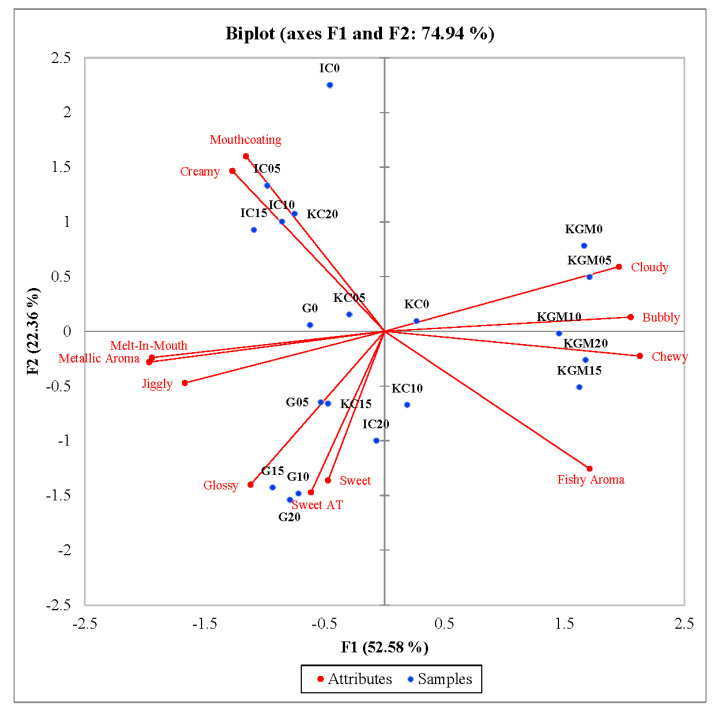
Principal components analysis biplot of Factors 1 and 2 by the correlation matrix of mean 12 sensory attributes across the 20 model confectionary gel samples based on Gel Type. AT = aftertaste, G = gelatin, IC = ι-carrageenan, KC = κ-carrageenan, KGM = konjac glucomannan.

**Figure 2 gels-08-00644-f002:**
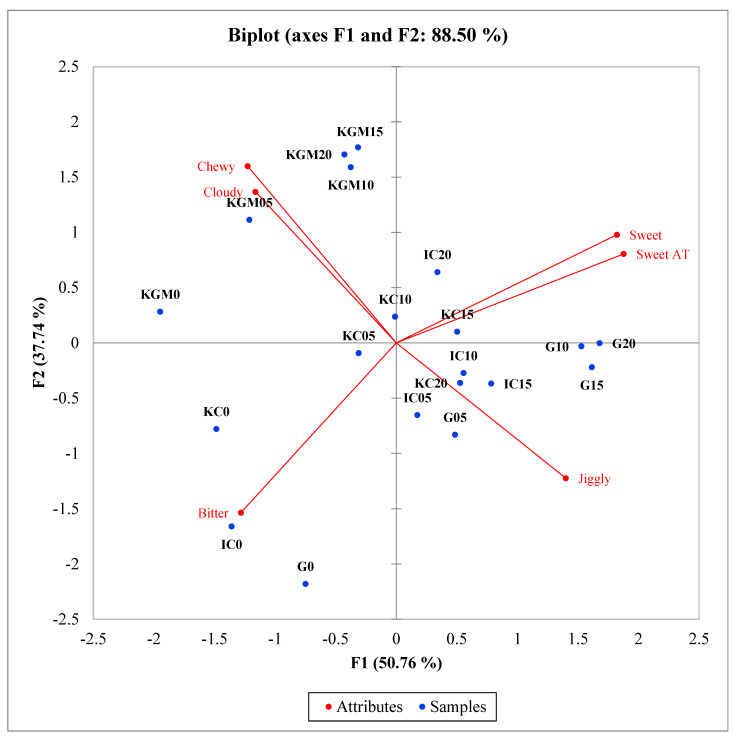
Principal components analysis biplot of Factors 1 and 2 by the correlation matrix of mean six sensory attributes across the 20 model confectionary gel samples based on sugar concentration. AT = aftertaste, G = gelatin, IC = ι-carrageenan, KC = κ-carrageenan, KGM = konjac glucomannan.

**Figure 3 gels-08-00644-f003:**
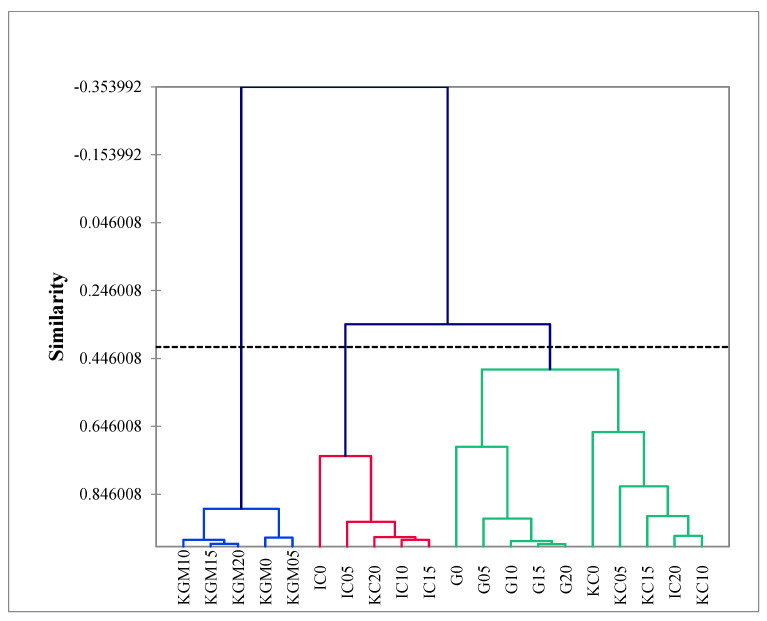
Dendrogram from agglomerative hierarchical cluster analysis (unweighted pair-group average method) of 20 model confectionary gel samples by sensory attributes based on gel type. Similarity based on Pearson correlation coefficient. G = gelatin, IC = ι-carrageenan, KC = κ-carrageenan, KGM = konjac glucomannan.

**Figure 4 gels-08-00644-f004:**
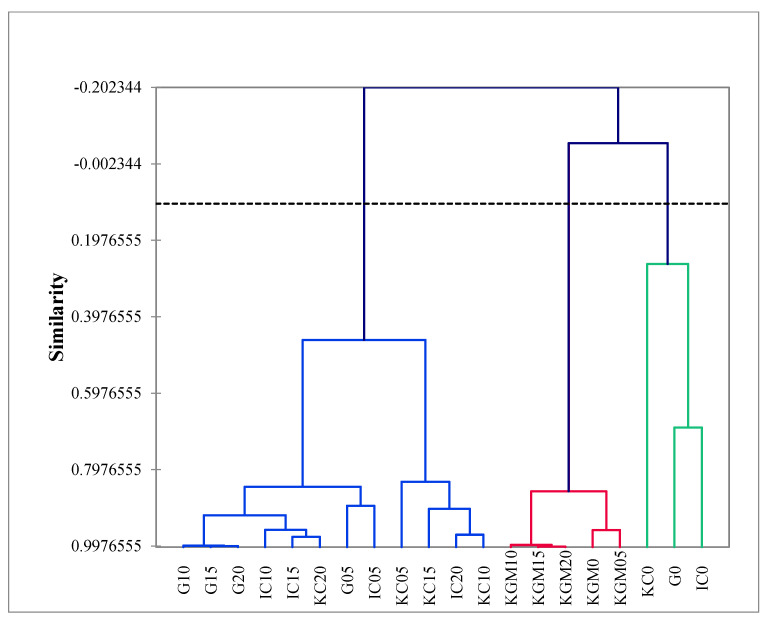
Dendrogram from agglomerative hierarchical cluster analysis (unweighted pair–group average method) of 20 model confectionary gel samples by sensory attributes based on sugar concentration. Similarity based on Pearson correlation coefficient. G = gelatin, IC = ι-carrageenan, KC = κ-carrageenan, KGM = konjac glucomannan.

**Figure 5 gels-08-00644-f005:**
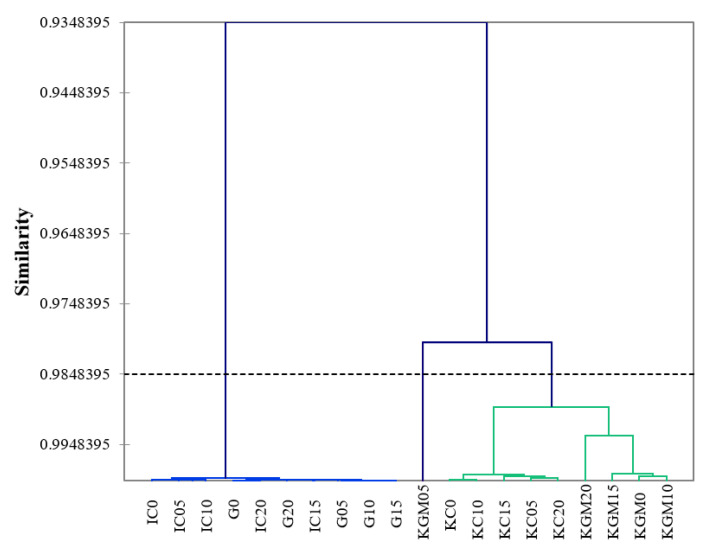
Dendrogram from agglomerative hierarchical cluster analysis (unweighted pair-group average method) of 20 model confectionary gel samples by instrumental measurements. Similarity based on Pearson correlation coefficient. G = gelatin, IC = ι-carrageenan, KC = κ-carrageenan, KGM = konjac glucomannan.

**Figure 6 gels-08-00644-f006:**
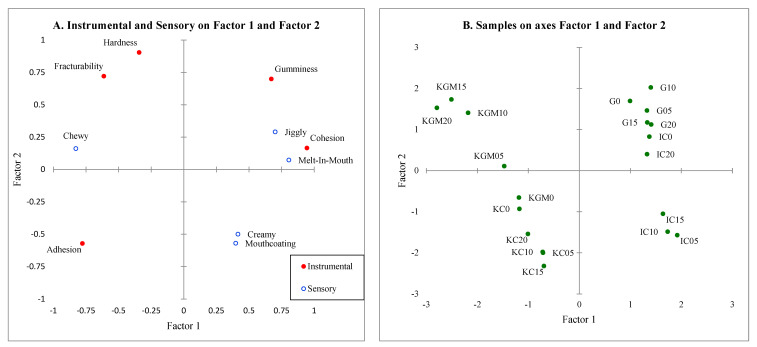
Partial least squares regression biplot for Factors 1 and 2 for Instrumental Parameters (x-variable) and Sensory Texture Attributes (y-variable) on 20 model confectionary gels: (**A**) correlations on Factor 1 and Factor 2; (**B**) observations on axes Factors 1 and 2. G = gelatin, IC = ι-carrageenan, KC = κ-carrageenan, KGM = konjac glucomannan.

**Table 1 gels-08-00644-t001:** Model confectionary gel attributes, definitions, reference brand, reference preparation and reference intensity developed by descriptive analysis panel. Reference intensities were iteratively calculated as the average intensity rating of the group using 16-point category scale from 0 to 15.

Modality	Term	Definition	Exact Reference Brand	Reference Preparation	Final Reference Intensity
Appearance	Cloudy	Difficulty to see through the sample	Minute Maid Lemonade (The Coca-Cola Company Inc., Atlanta, GA, USA)	20 mL lemonade of diluted 1:1 with filtered water (in a 29.5 mL cup)	7.1
	Glossy	Ability to reflect light on the surface	Meijer Light Corn Syrup (Meijer, Inc., Grand Rapids, MI, USA)	10 g of corn syrup (in 29.5 mL cup)	11.0
	Bubbly	Amount of air pockets in the sample	Meijer After Sun Aloe Gel (Meijer, Inc., Grand Rapids, MI, USA)	10 g of aloe vera shaken before pumped (into 29.5 mL cup)	7.1
Aroma	Fishy	Aroma associated with aquariums, fish tanks, or fish	Bumble Bee Chunk Light Tuna (Bumble Bee Foods, LLC, San Diego, CA, USA)	Liquid in tuna can (25 g) was taken and diluted with 1 cup of filtered water (20 mL in 29.5 mL cup)	11.0
	Metallic	Aroma associated with metals	USA	Penny is held in hand while smelling (taken out of the 29.5 mL cup)	7.7
Taste	Sweet	Taste associated with sugar	Meijer Pure Granulated Sugar (Meijer, Inc., Grand Rapids, MI, USA)	3.5% *w*/*v* sucrose in filtered water (20 mL in 29.5 mL cup)	9.7
	Bitter	Taste associated with caffeine	Sigma-Aldrich 99% Caffeine (Merck KGaA, Darmstadt, Germany)	0.1% *w*/*v* caffeine in filtered water (20 mL in 29.5 mL cup)	7.8
Aftertaste	Sweet	Aftertaste associated with sugar	Meijer Pure Granulated Sugar (Meijer, Inc., Grand Rapids, MI, USA)	3.5% *w*/*v* sucrose in filtered water (20 mL in 29.5 mL cup)	8.3
Texture	Jiggly	Degree the sample wobbles when the container is slightly tapped with the finger	Jell-O Raspberry Flavor Gelatin Dessert (The Kraft Heinz Company, Chicago, IL, USA)	Jell-O prepared according to package (20 g in 29.5 mL cup)	11.6
	Creamy	Rich and smooth texture when eaten	Snack Pack Chocolate Pudding (Conagra Brands, Inc., Chicago, IL, USA)	20 g of chocolate pudding (in 29.5 mL cup)	10.9
	Chewy	Resistance when chewing	Welch’s Mixed Fruit Fruit Snacks (Welch Foods Inc., Concord, MA, USA)	Two strawberry gummies (in 29.5 mL cup)	9.3
	Melt-In-Mouth	Dissolves or falls apart in mouth without force or resistance	Meijer Whipped Cream (Meijer, Inc., Grand Rapids, MI, USA)	Whipped cream in 29.5 mL cup filled completely	11.2
	Mouthcoating	Leaving a residue or film in mouth after the food leaves	Meijer Heavy Whipping Cream (Meijer, Inc., Grand Rapids, MI, USA)	10 mL of whipping cream (in 29.5 mL cup)	11.0

**Table 2 gels-08-00644-t002:** Analysis of variance of 13 sensory attributes rated for model confectionary gels samples. F-ratios are shown for the sources of variations and interactions. *, **, *** indicated significance at *p* < 0.05, *p* < 0.01, *p* < 0.001. AT = aftertaste.

Attributes	Judge (J)	Gel Type (GT)	Sugar Concentration (SC)	Replication (R)	J X GT	J X SC	J X R	GT X SC
Cloudy	12.10 ***	296.79 ***	6.28 ***	0.44	12.53 ***	2.62 ***	0.52	26.92 ***
Glossy	54.97 ***	13.20 ***	2.31	1.74	10.31 ***	1.97 **	0.92	22.60 ***
Bubbly	22.06 ***	192.66 ***	2.02	1.38	18.59 ***	0.94	1.25	3.26 ***
Fishy Aroma	15.55 ***	11.63 ***	1.40	0.53	6.30 ***	1.74 *	1.61	6.86 ***
Metallic Aroma	34.80 ***	5.40 **	2.00	0.04	5.59 ***	1.40	1.16	0.76
Sweet	36.27 ***	7.13 ***	32.16 ***	0.03	4.05 ***	8.88 ***	3.02 **	3.19 ***
Bitter	45.07 ***	1.71	14.40 ***	2.98	4.27 ***	4.73 ***	1.97 *	2.29 **
Sweet AT	66.20 ***	14.98 ***	26.75 ***	0.23	3.34 ***	9.16 ***	1.56	5.19 ***
Jiggly	34.42 ***	84.37 ***	3.43*	0.36	5.09 ***	2.01 **	1.06	20.96 ***
Creamy	45.83 ***	10.03 ***	2.07	0.34	8.16 ***	2.48 ***	0.60	19.35 ***
Chewy	15.96 ***	16.39 ***	3.08*	0.35	15.46 ***	1.23	0.37	14.07 ***
Melt-in-Mouth	66.40 ***	27.96 ***	1.19	0.13	6.36 ***	1.31	0.92	7.82 ***
Mouthcoating	33.68 ***	45.10 ***	1.30	0.50	5.25 ***	1.87 **	1.10	16.79 ***

**Table 3 gels-08-00644-t003:** Correlation matrix of the mean sensory textural ratings and instrumental measurements across samples. Values in bold are different from 0 with a significance level α = 0.05. TPA = texture profile analysis.

Variables	Hardness (TPA)	Fracturability (TPA)	Gumminess (TPA)	Adhesion (TPA)	Cohesion (TPA)	Jiggly	Melt-in-Mouth	Chewy	Creamy	Mouthcoating
Hardness (TPA)	**1**									
Fracturability (TPA)	**0.88**	**1**								
Gumminess (TPA)	0.44	0.08	**1**							
Adhesion (TPA)	−0.25	0.05	**−0.96**	**1**						
Cohesion (TPA)	−0.19	**−0.45**	**0.78**	**−0.87**	**1**					
Jiggly	0.07	−0.31	**0.73**	**−0.73**	**0.71**	**1**				
Melt-In-Mouth	−0.18	**−0.54**	**0.65**	**−0.68**	**0.77**	**0.94**	**1**			
Chewy	0.43	**0.71**	**−0.48**	**0.58**	**−0.76**	**−0.83**	**−0.94**	**1**		
Creamy	**−0.66**	**−0.56**	−0.12	−0.07	0.39	0.27	0.41	**−0.63**	**1**	
Mouthcoating	**−0.72**	**−0.60**	−0.19	−0.02	0.36	0.07	0.24	**−0.52**	**0.94**	**1**

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
