# Peer review of "Characterization of Sugar Reduction in Model Confectionary Gels Using Descriptive Analysis"

_gels, 2022, doi:10.3390/gels8100644_

Round 1

Reviewer 1 Report

A very interesting article on sugar reduction in food and how it affects sensory properties. 

The article is very well thought out and developed, which together with the intrinsic interest of the subject matter makes me find it optimized for publication in this prestigious journal.

Author Response

Thank you for taking the time to review our manuscript.

Reviewer 2 Report

In this study, the authors investigated the effects of sugar reduction on model confectionary gels made from various gelling agents, and correlated and compared sensory and instrumental measurements to determine which attributes best describe the model confectionary gels. This study highlights the importance of sensory perception in products. However, this work is not appropriately designed and thus the conclusions are not strongly supported by the results. The manuscript can be reconsidered after major revision:

1.       In the Introduction, the authors indicated that less work has been done regarding the aspect of sugar reduction, especially in the area of confectionary gels. However, the authors should provide more information to define the significance of this work.

2.       It is important to clearly describe how the samples were prepared, especially for as many as 20 samples were used in this study. For example, what are the sugar concentrations in the samples? It needs to be mentioned in the section of sample preparation.

3.       Line 27. (Pacheco et al., 2020) and Page 9 Suebsaen et al. (2019) cannot be found in the references.

4.       Table 1 is not important, and could be moved to the supplementary materials.  

Author Response

In this study, the authors investigated the effects of sugar reduction on model confectionary gels made from various gelling agents and correlated and compared sensory and instrumental measurements to determine which attributes best describe the model confectionary gels. This study highlights the importance of sensory perception in products. However, this work is not appropriately designed and thus the conclusions are not strongly supported by the results. The manuscript can be reconsidered after major revision:

  • Thank you for your review. Your comments have been addressed below and in the manuscript. More has been added to address the design of the experiment and the significance.
  1. In the Introduction, the authors indicated that less work has been done regarding the aspect of sugar reduction, especially in the area of confectionary gels. However, the authors should provide more information to define the significance of this work.
  • More information has been added to define the significance of the study. Line 69-76
  • Line 69-76: Developing a correlation between sensory and instrumental results is critical for product development and the success of sugar reduction in products. The consumer or sensory perception of a product is often the final determinant of hedonic results. This research works to characterize and highlight the changes in sensory perception that occur during sugar reduction in gel-like products and correlate that to instrumental data. This sensory and correlation data can be used as reference for future work when initially attempting to reduce sugar in gel-like products.
  1. It is important to clearly describe how the samples were prepared, especially for as many as 20 samples were used in this study. For example, what are the sugar concentrations in the samples? It needs to be mentioned in the section of sample preparation.
  • Gelling agent concentrations were added to sample preparation section. Line 90,93, and 96.
  • Rationale for gelling agent concentration and sugar concentrations were added to the sample preparation section. Line 102-106
  • LINE 102 to 106: Gelling agent concentrations were selected which maintain a consistent initial hardness from texture profile analysis data in samples with no sugar. If sugar was added to samples, it was added simultaneously with the gelling agent. Sugar concentrations were 0, 5, 10, 15, and 20% w/v for each gelling agent. Overall, 20 samples were tested.
  1. Line 27. (Pacheco et al., 2020) and Page 9 Suebsaen et al. (2019) cannot be found in the references.
    • References in text were added to the reference section
  1. Table 1 is not important, and could be moved to the supplementary materials.  
  • Table 1 was removed from the main portion of the text and added to the appendix.
  • The subsequent table numbering was adjusted.

Reviewer 3 Report

The authors did not provide a powerful discussion of the results with previous literature since the authors only focus on describing their results. The scientific quality of the manuscript is insufficient for publication in its current form. Specific questions and points requiring attention are itemized below.

Review comments: “Overconsumption of added sugar and high sugar products can lead to numerous health problems that result from the consequential weight gain, such as cardiovascular disease”. Describe more diseases….

Review comments: The authors must revise the format of the References in the text.

Review comments: “However, by modifying a product in such a way comes the challenge of maintaining both its functional and sensory aspects”. The author must describe more in detail what functional and sensory aspects are refered….

Review comments: It is still difficult to find the novelty of the work concerning what has already been published. What is the difference between what is published with what the authors want to publish? It is not clear. The authors must describe these differences in the introduction section.

Reviewer’s comment: Reviewer´s comment: What is the innovation of this paper? What is new in this work? It is not clear.

 Review comments: The introduction was poorly written. It requires additional information on previous attempts when similar materials were used and what were the results.

Reviewer’s comment: The results and discussion sections are poor. More comparisons with previous literature should be discussed. The authors can discuss with the publication: DOI 10.3390/molecules27154902.

Reviewer’s comment: Describe the function of each ingredient in the formulation.

Author Response

The authors did not provide a powerful discussion of the results with previous literature since the authors only focus on describing their results. The scientific quality of the manuscript is insufficient for publication in its current form. Specific questions and points requiring attention are itemized below.

  • Thank you for the review. The comments have been addressed below and in the manuscript. More has been added to provide a powerful discussion.

Review comments: “Overconsumption of added sugar and high sugar products can lead to numerous health problems that result from the consequential weight gain, such as cardiovascular disease”. Describe more diseases….

  • Type 2 diabetes mellitus is mentioned in the sentence. These two health problems cover most associated with weight gain. More could be added if necessary. Line 27

Review comments: The authors must revise the format of the References in the text.

  • References have been formatted in text and in the reference section.

Review comments: “However, by modifying a product in such a way comes the challenge of maintaining both its functional and sensory aspects”. The author must describe more in detail what functional and sensory aspects are referred….

  • Functional aspects are detailed in the sentence following the mentioned. More explanation was added to describe the changes in sensory aspects. Line 37-39-XX and Line 41-42
  • Line 37-39: Sensory perception of reduced sugar products is also affected. For example, sugar often provides a different taste profile than sugar alternatives, which often have licorice or metallic aftertastes.
  • Line 41-42: While there are many roles that sugar holds in product development, one of the main effects of sugar reduction are modifications in mouthfeel and texture, which alter both the instrumental and sensory results [9], [10].

Review comments: It is still difficult to find the novelty of the work concerning what has already been published. What is the difference between what is published with what the authors want to publish? It is not clear. The authors must describe these differences in the introduction section.

  • More information has been added to define the significance of the study. Line 64-66 and Line 69-76
  • Line 64-66: they usually lack a sensory component, and because the correlation between instrumental and sensory data is dependent on the matrix, more research is required to characterize sugar reduction in individual food products, such as gels.
  • Line 69-76: Developing a correlation between sensory and instrumental results is critical for product development and the success of sugar reduction in products. The consumer or sensory perception of a product is often the final determinant of hedonic results. This research works to characterize and highlight the changes in sensory perception that occur during sugar reduction in gel-like products and correlate that to instrumental data. This sensory and correlation data can be used as reference for future work when initially attempting to reduce sugar in gel-like products.

Reviewer’s comment: Reviewer´s comment: What is the innovation of this paper? What is new in this work? It is not clear.

  • More information was added in the introduction to explain the innovation. This work adds to literature describing the correlation between instrumental and sensory data for various food product categories. The objective has been edited to reflect the innovation of the paper. Line 77-78
  • Line 77-78: The objectives of this research were to investigate the effects of sugar reduction on model confectionary gels made from various gelling agents, some which are under characterized in the literature, and to correlate and compare sensory and instrumental measurements to determine which attributes best describe the model confectionary gels.

 Review comments: The introduction was poorly written. It requires additional information on previous attempts when similar materials were used and what were the results.

  • Additional information on previous research was added to the introduction. However, the results from them were usually focused on processing method, source of gelling agent, or just instrumental data. As the focus of the paper is on the correlation between the two, details of results were not added in the introduction. More information of previous or similar attempts were added to the discussion.
  • Line 59-66: Other works have utilized this method to characterize gel systems, such as Yusof et al., 2019 [13], Chandra et al., 2015 [14]and Rahman M et al., 2009 [15]. In these studies, TPA was the main technique to characterize changes in texture based on different processing methods and gelling agent sources, specifically in gelatin. While there are studies which instrumentally characterize the changes in gel systems when sugar reduction occurs [16]–[18], they usually lack a sensory component, and because the correlation between instrumental and sensory data is dependent on the matrix, more research is required to characterize sugar reduction in individual food products, such as gels.

Reviewer’s comment: The results and discussion sections are poor. More comparisons with previous literature should be discussed. The authors can discuss with the publication: DOI 10.3390/molecules27154902.

  • Thank you for the suggestion. Some comparisons to previous literature have been added. However, the recommended paper only reports instrumental data (TPA) and not sensory. The focus of the research paper is the comparison of instrumental and sensory.
  • PAGE 9; Line 17-20 : Similar results were found in studies which attempted to replace sugar with alternative sweeteners in jam-like products. Akesowan and Choonhahirun (2019) found that “harder” gels were perceived as less sweet, due to the limited flavor release.
  • PAGE 11; Line 64-66: which has been exhibited in the literature (Kappes, 2006). In work done by Kappes et al (2006), panelists were able to detect mouthfeel differences between diet and regular carbonated beverages, despite the instrumental results being minute.
  • PAGE 12; Line 93-96: This aligns with other studies, such as Garcia Loredo et al (2011), which looked at the correlation between instrumental and sensory ratings. The researchers found that hardness and fracturability had higher correlation with their TPA equivalents.

Reviewer’s comment: Describe the function of each ingredient in the formulation.

  • Function added in raw materials and sample preparation section Line 84
  • Function of calcium hydroxide is in LINE 86
  • Sugar is just an added ingredient and does not serve a specified function, so no description was added.

Round 2

Reviewer 2 Report

The authors have answered all my questions and improved the quality of manuscript according to the comments and suggestions from the reviewers. To my opinion, this revised review can be accepted.

Reviewer 3 Report

The manuscript can be accepted